# Apophyseal Avulsion of the Rectus Femoris Tendon Origin in Adolescent Soccer Players

**DOI:** 10.3390/children9071016

**Published:** 2022-07-08

**Authors:** Hanneke Weel, A. J. Peter Joosten, Christiaan J. A. van Bergen

**Affiliations:** 1Bergman Clinics, Department of Orthopedics Arnhem, Mr. E.N. van Kleffensstraat 14, 6842 CV Arnhem, The Netherlands; h.weel@bergmanclinics.nl; 2Department of Orthopedic Surgery, Amphia Hospital, 4800 RK Breda, The Netherlands; ajoosten@amphia.nl

**Keywords:** apophysial avulsion, rectus femoris tendon, adolescent, sport

## Abstract

Apophyseal avulsions of the rectus femorus tendon (RFT) at the anterior inferior iliac spine (AIIS) can occur in adolescents, often while performing soccer. Patient-reported outcomes (PROMs) and time to return to sport of these patients are relatively unknown. Therefore, the aim of this study was to assess the PROMs and return to sports of patients with AIIS avulsions and compare the results with those reported in the literature. This is a case series of seven consecutive patients presenting at our hospital between 2018 and 2020 with an apophyseal avulsion of the RFT from the AIIS. The patients were assessed with use of the WOMAC and Tegner scores and return to sports was evaluated. All patients were male soccer players (median age 13 years; range, 12–17). They were all initially treated non-operatively. One of the patients subsequently needed excision surgery of a heterotopic ossification because of non-transient hip impingement. All other patients recovered after a period of relative rest. Median time to return to sports was 2.5 months (range, 2–3). At a median follow-up of 33 months (range, 18–45), the WOMAC (median, 100; range, 91–100) and Tegner scores (median, 9; range, 5–9) were high. In accordance with the existing literature, most patients with apophyseal avulsions of the AIIS recover well with non-operative treatment. However, the avulsion can lead to hip impingement due to heterotopic ossifications possibly needing surgical excision. Sport resumption is achievable after 2–3 months, and patient-reported outcomes are highly satisfactory in the long term.

## 1. Introduction

In the paediatric pelvis, the apophyseal plate is a biomechanical weak spot [1] because the cartilaginous growth plate fails in tension before the musculotendinous unit does [2]. Apophyses are at risk to avulsion fractures in adolescents, especially in athletes due to strong contraction forces of the attached muscles. Pelvic avulsions are often seen in adolescent athletes, with the avulsion at the anterior inferior iliac spine (AIIS) counting for 22–49% of the pelvic avulsions, followed by the anterior superior iliac spine with 20–30% [3,4,5,6]. The direct head of the rectus femoris tendon (RFT) originates from the AIIS, proximal to the hip joint, whereas the reflected head originates from the anterior acetabular ridge and anterior hip joint capsule, which is rarely affected by avulsion injuries. The apophysis is especially at risk of injury between formation of the secondary growth centre and its closure. For the AIIS, this period lays between ages 13.6 and 16.3 years in boys and between 14 and 14.9 in girls [2]. Avulsion injury of the AIIS typically occurs with eccentric force at the hip, such as seen in sprinting and kicking a ball. Boys are more often affected than girls [4], with up to 70% of the affected adolescents performing ball sports [3], but also track and field [3,4] and rarely skiing [7].

Patients with AIIS injuries may describe feeling or hearing a “pop” at the time of injury. This is reported to be present in 33% [5]. Swelling and ecchymosis may be present. According to Müller et al. physical examination demonstrates tenderness at the AIIS on palpation (in 98%), weakness of muscles (85%), pain during motion (47%), and sometimes a notable limping (23%) [5]. Pain and weakness with hip flexion, knee extension, or a resisted straight leg raise also may be present [5].

Displacement of the apophysis is thought to be restricted by a relatively thick periosteum. When the avulsion is not more than 15 [3] or 20 mm [8] displaced, the treatment of choice is conservative. Conservative treatment mostly starts with rest, builds up with progressively regaining motion until allowance of weight bearing, then starting training muscle strength until return to sports, as first described in detail by Metzmaker and Pappas [9]. Although non-surgical treatment usually leads to high success rates in the short term [3,4,8,9,10], patients with AIIS avulsions are 4.5 times more likely to experience future hip pain beyond 3 months compared with other pelvic avulsions [4]. Complication rates are reported to be high (64%) and similarly distributed over both nonoperative and operative treatments, with non-union and heterotopic ossifications mostly reported [3]. However, return to sports and long-term patient-reported outcomes are rarely reported in the literature. Therefore, the aim of this study was to assess return to sports and long-term patient-reported outcomes of adolescents with avulsion fractures of the AIIS.

## 2. Materials and Methods

### 2.1. Patients

This retrospective case series included all consecutive patients presenting to the authors’ outpatient clinic of the orthopaedic department of a large teaching hospital (Amphia Hospital, Breda, the Netherlands) from January 2018 until the end of 2020. Inclusion criteria were adolescents, defined by the World Health Organisation (WHO) as patients between 10 and 19 years of age, minimum follow-up of 1 year, and confirmation of the diagnosis on radiography of the pelvis. There were no exclusion criteria.

### 2.2. Assessment

Collected injury data included: side of injury, type and level of sports, trauma mechanism (if applicable), age, gender, any signs of prodromal symptoms, previous treatment, and underlying illnesses/previous medical history. At a minimum of 12 months after presentation, patients were contacted. After informed consent of patient and caretaker was obtained, they completed two patient-reported outcome measures (PROMs). Because of the retrospective design of the study, consisting of data without burden for the involved subjects, the approval of a medical ethics committee was not required.

### 2.3. WOMAC

The Western Ontario and McMaster Universities Osteoarthritis Index (WOMAC) was used to assess the patients at follow-up. The WOMAC evaluates three dimensions: pain, stiffness, and physical function with 5, 2, and 17 questions, respectively. The Likert version of the WOMAC is rated on an ordinal scale of 0–4, with lower scores indicating lower levels of symptoms or physical disability. Each subscale is summated to a maximum score of 20, 8, and 68, respectively. There is also an index score or global score, which is most commonly calculated by summating the scores for the three subscales. The questionnaire has been validated in Dutch [11], self-administered and takes 5–10 min to complete [12]. It is the only anatomic-specific paediatric sports PROM of the hip available, validated in a Dutch population aged 12–35 years [13].

### 2.4. Tegner Score

The Tegner score is an activity score, filled out by the patient. It rates the physical intensity of performed work and/or sports, scoring from 0 to 10. A higher score corresponds with a higher activity level [14].

### 2.5. Data Analysis

The data were processed descriptively. Patient demographics were summarised. Because of the small number of patients, median and range were used to describe the continuous data. Data analysis was performed using IBM SPSS Statistics Version 26.0 (IBM, Armonk, NY, USA).

## 3. Results

Seven patients met the inclusion criteria (Table 1), and six could be included at the final follow-up (Table 2). All patients were adolescent, with a median age of 13 years (range, 12–17). They were all male and soccer players. All were without relevant medical history, without signs of hyperlaxity and with a negative family history for orthopaedic problems. The median displacement of the apophysis from the AIIS as measured on the radiographs was 8 mm (range, 0–15 mm).

### 3.1. Treatment

Initially, all patients were treated conservatively. The treatment consisted of relative rest for 6 weeks, followed by a personalised program with support of a physiotherapist, containing gradual progress of activity until return to sports.

One patient required surgery, after initial conservative treatment failed (case 7 in Table 1 and Table 2). Despite intensifying physiotherapy, no progression was seen. When 2.5 months passed by after presentation, he still had symptoms, consisting of pain and limitations of the hip, which worsened over time. New radiographs and a computed tomography (CT) scan were made and showed an ossified non-union avulsion of the rectus femoris tendon (RFT) from the AIIS with heterotopic ossification (Figure 1 and Figure 2). The heterotopic ossification caused mechanical symptoms during sports and his work as a plumber. Therefore, surgery was performed to remove the ossification.

During surgery, the ossification was removed using the Smith–Petersen approach. Through the interval between the tensor fascia lata and the sartorius muscles, the RFT was split and the ossification removed with an osteotome. The 3.5 by 2 by 2 cm ossification was sent to the pathologist for analysis. Then, the hip was taken through range of motion, showing no impingement. The RFT split and the fascia were closed, leaving the lateral femoral cutaneous nerve intact.

Postoperatively, the patient was allowed to partially bear weight with crutches for 4 weeks and started training with a physiotherapist. The first 2 weeks he was prescribed NSAIDs to prevent recurrence of the ossification. No peak load or kicking was allowed for 10 weeks. The pathological exam showed a benign calcification. At 6 and 12 weeks, the patient was pain free and had a full range of motion of the hip. Radiography at 6 weeks postoperatively did not show a remnant or recurrence (Figure 3). At 12 weeks, the patient was running and working as a plumber again without symptoms. No complications were observed. At final follow up after 31 months, he had quit soccer due to loss of interest, but was still without any complains.

### 3.2. Outcomes

All patients returned to soccer, except for one who lost interest (see Table 1). The median time to return to sports was 2.5 (range, 2–3) months (see Table 1).

In total, 6 out of 7 patients were included at the final follow-up of a median 33 months (range, 18–45). In all patients, including the one who was treated surgically, WOMAC-scores were high to perfect, with a median of 100 (range, 91–100). Likewise, Tegner scores were high, with a median of 9 (range, 5–9), indicating competitive soccer (Table 2).

## 4. Discussion

In this case series, we present seven soccer players with an AIIS avulsion. Six of them could be successfully treated with a conservative physiotherapy program. In one of them, conservative treatment failed because of mechanical impingement due to a heterotopic ossification. This patient was successfully treated with surgical resection of the ossification.

To our knowledge, we were the first to assess long-term PROMs in adolescents with avulsion fractures of the AIIS. In all patients, all domains of the WOMAC score as well as the total WOMAC score were high, indicating good and pain-free hip function. The Tegner scores, representing level of sports and physical work activities, were also reported to be high.

In the literature, only limited reported cases of surgically treated AIIS avulsions are reported [15,16], most likely due to high success rates with conservative treatment, being reported between 75 and 100% [4,9]. Nevertheless, the AIIS avulsion are described to be 4.47 times more likely to experience future hip pain beyond 3 months compared with other pelvic avulsions [4]. A variety of reasons for these ongoing symptoms are named, such as non-union, hip impingement, heterotopic ossifications, re-fractures, and tendinopathies. It has also been suggested that there may be an association between avulsion of the reflected head of the RFT and labral injuries; in a retrospective study, 2 out of 9 RFT avulsions had labral lesions on magnetic resonance (MR) arthrogram [17].

In our series, one of the patients required surgery, because of impingement and obstructive symptoms due to heterotopic ossification. Lambrechts et al. also reported that secondary surgery can be needed in case of a heterotopic ossification [18]. Other cases show that swelling and pain of an AIIS avulsion can sometimes be mistaken for malignancies, resulting into excision surgery [19,20,21,22]. The local tenderness combined with an exostosis on imaging studies may mimic a (pseudo-) tumour.

Recently, it was suggested that adolescent sport players are at risk of rectus femoris avulsion fractures at the AIIS when there is a lack of abdominal muscle strength [23]. We do not have data on this hypothesis, but 2 out of 7 patients reported to have experienced symptoms, previous to the injury. One of them even had an avulsion at both sides. Theoretically, this could be a very cautious suggestion of too much stress on the apophysis in the growing skeleton. From other apophysitis, for example, the little league elbow, we do know that overload is a substantial risk factor [24].

All patients returned to soccer, except for one who lost interest, with a median time to return to sports being 2.5 (range, 2–3) months. In all patients, including the one who was treated surgically, WOMAC-scores were high to perfect, with a median of 100 (range, 91–100). Likewise, Tegner scores were high, with a median of 9 (range, 5–9). A review by Caderazzi [25] evaluated the return to sport rate in 86 patients; 90% of the conservative group and 95% of the surgical patients returned to sports at follow-up, being comparable to our series. The complication rate in the conservative group of Caderazzi et al. was 18%, compared to 22% in the surgical group. The rate of non-unions was lower in the surgical group (0%) than in the conservative group (2.5%), whereas there were more heterotopic ossifications in patients treated surgically (9% vs. 1.8%). In our case series, only one patient required secondary surgery. There were no further complications detected in the patients treated conservatively, nor in the surgically treated patient.

Plain radiographs are included in the initial diagnostic workup, with anteroposterior and frog-leg lateral views of the pelvis and hip. When negative, but with persistent suspicion of an avulsion, CT or MR may assist if the diagnosis is unclear on initial radiographic evaluation [26]. These additional images can also visualise more clearly the amount of displacement, possibly influencing choice of treatment. In our series, 4 out of 7 needed additional work-up, mostly to more precisely measure displacement and determine treatment.

This study has limitations. The major limitation is the relatively small series, and non-standard follow-up times. There was one missing respondent in the long-term follow-up evaluation.

## 5. Conclusions

In adolescent soccer players, pain in the groin region should be taken seriously, because the apophyseal plate is a biomechanical weak spot. In this case series, we presented seven cases, of whom six had good results with physiotherapy-guided conservative treatment. One patient required surgical removal of a heterotopic ossification. Sports were resumed after 2–3 months. Long-term follow-up showed high scores on WOMAC and Tegner, in both the surgically and conservatively treated patients, indicating good and pain-free functioning of the hip and high levels of physical activity.

## Figures and Tables

**Figure 1 children-09-01016-f001:**
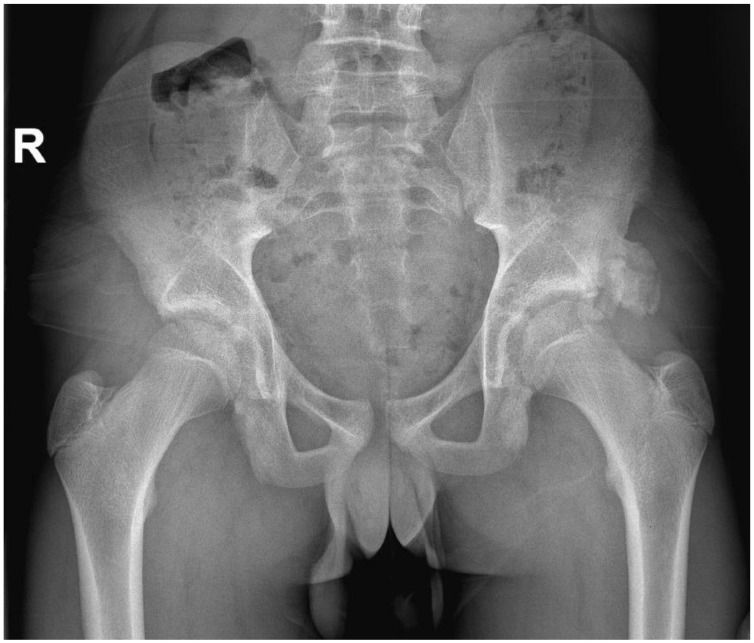
X-ray of the pelvis 2 months after an avulsion of the origin of the left rectus femoris tendon, showing a big osseous calcification on the anterior inferior iliac spine. (Only relevant findings are described).

**Figure 2 children-09-01016-f002:**
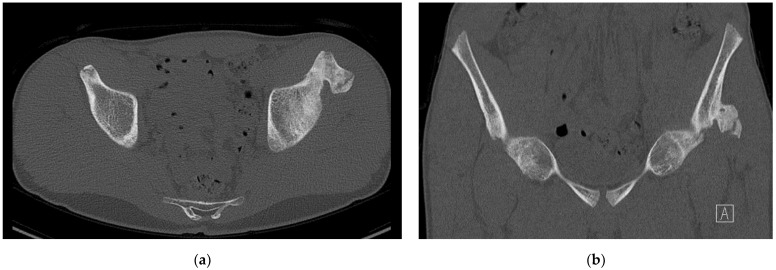
Computed tomography scan with an axial (**a**) and a coronal (**b**) view of the osseous calcification on the left anterior inferior iliac spine, showing a heterotopic ossification of the origin of the rectus femoris tendon.

**Figure 3 children-09-01016-f003:**
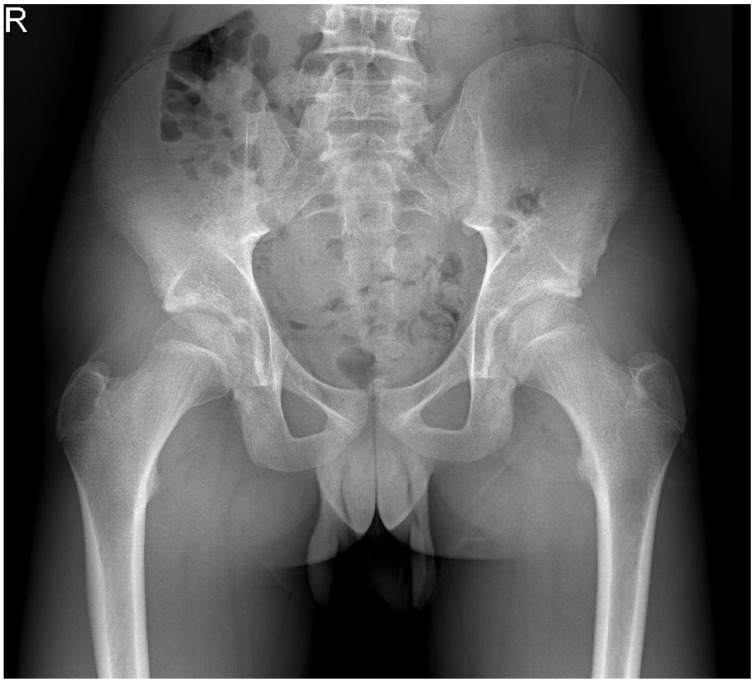
Plain pelvic radiograph 6 weeks after surgical removal of the heterotopic ossification.

**Table 1 children-09-01016-t001:** Patient characteristics at presentation.

Case	Age	Gender	Side	Previous Symptoms	Trauma Mechanism	Diagnosis	Displacement from AIIS	Time to RtS (Months)
1	12	Male	R	None	Kicking a ball	X	5 mm	NA
2	13	Male	R	None	Kicking a ball	X/CT	15 mm	3
3	13	Male	R	None	Kicking a ball	X/US	0 mm	2
4	13	Male	L + R	Yes	No trauma	X	0 mm	Unknown *
5	15	Male	R	None	Kicking a ball	X	8 mm	2
6	16	Male	L	None	Fall on knee	X/MRI	10 mm	3
7	17	Male	L	Yes	Jump	X/CT	10 mm	3

AIIS: anterior inferior iliac spine; CT: computed tomography; L: left; MRI: magnetic resonance imaging; NA: not applicable (quit soccer because lost interest); R: right; RtS: return to sport; US: ultrasound; X: X-ray; * already recovered from injury at presentation.

**Table 2 children-09-01016-t002:** Patient-reported outcomes at final follow-up.

Case	WOMAC Pain	WOMAC Joint	WOMAC ph.F	WOMAC Total	Tegner Score	FU (m)
1	100	100	100	100	6	45
2	95	100	88	91	9	18
3	100	100	100	100	9	43
4	100	100	100	100	9	34
6	100	100	100	100	9	20
7	100	100	100	100	5	31

WOMAC: Western Ontario and McMaster Universities Osteoarthritis Index; ph.F: physical functioning; FU (m): follow-up (months).

## Data Availability

Not applicable.

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
