# Peer review of "Apophyseal Avulsion of the Rectus Femoris Tendon Origin in Adolescent Soccer Players"

_children, 2022, doi:10.3390/children9071016_

Round 1
Reviewer 1 Report
This study is a brief and well-presented case series describing long term patient-reported outcomes of apophyseal avulsion at the anterior inferior iliac spine.
All patients were adolescent male soccer players. All were initially treated conservatively. The patient-reported outcomes were mostly good, with exception of one of seven patients who required surgical removal of an ossification that had led to hip impingement. The study is described well and contributes to knowledge in the field. The main limitation is the low case number: although this is a relatively common injury, only seven cases are described. Expanding the study to other participating clinics would have improved the case numbers and thereby increased the statistical power, and impact, of this study.
Other comments/suggestions:
In the abstract results/conclusion, please refer to the comparison with other studies (as this comparison is part of the study aim).
In the methods section, please describe the clinic, setting and inclusion process. What was the exact location/hospital, what were the exact dates for this study? How were patients (and/or parents) approached? How many (if any) did not agree to participate?
Please give details of the ethical approval for this study. Some specific patient details (age, sex, sport, injury type, occupation), combined, may allow for re-identification of the patient information.
Please replace the word ‘complaints’ with ‘symptoms’ (or other appropriate term) throughout the manuscript.
Reviewer 2 Report
Re- Apophyseal avulsion of the rectus femoris tendon origin in ad- 2 olescent soccer players
Line18 : One of the patients subsequently needed excision surgery of an heterotopic ossification because of non-transient hip impingement
Comment: This patient requires better clinical documentation.
Line 78: The Western Ontario and McMaster Universities Osteoarthritis Index (WOMAC) was used to assess the patients at follow-up.
Comment: Authors need to use additional scoring system such as Beighton score to confirm or rule out ligamentous hyperlaxity.
Table 1. Patient characteristics at presentation
Comment: The table is insufficient to cover the clinical phenotype. Heights of the patients? Family history of orthopedic abnormalities such as fractures, Slipped capital femoral epiphysis (SCFE), protrusio acetabluae, flat foot etc..?
Figure 1. X-ray of the pelvis 2 months after an avulsion of the origin of the left rectus femoris tendon, 123 showing a big osseous calcification on the anterior inferior iliac spine.
Comment: What I see in this AP pelvis radiograph: Bilateral and symmetrical sclerosis and dysplasia of the acetabulae, suspected spina bifida of the 5 th lumbar vertebra. Bilateral hypoplastic capital femoral epiphyses??
The current clinical documentation is insufficient
Round 2
Reviewer 2 Report
To Authors,
Assessing the ligaments is basic in children with avulsion fractures. AP pelvis radiograph shows acetabular sclerosis. Attached is your radiograph, please check.
